# Septin Defects Favour Symmetric Inheritance of the Budding Yeast Deceptive Courtship Memory

**DOI:** 10.3390/ijms24033003

**Published:** 2023-02-03

**Authors:** Fozia Akhtar, Bastien Brignola, Fabrice Caudron

**Affiliations:** 1School of Biological and Behavioural Sciences, Queen Mary University of London, London E1 4NS, UK; 2IGMM, Univ Montpellier, CNRS, 34293 Montpellier, France

**Keywords:** septin, mnemon, asymmetric cell division, memory, diffusion barrier, Whi3

## Abstract

Mnemons are prion-like elements that encode cellular memories of past cellular adaptations and do not spread to progenies during cell divisions. During the deceptive courtship in budding yeast, the Whi3 mnemon (Whi3^mnem^) condenses into a super-assembly to encode a mating pheromone refractory state established in the mother cell. Whi3^mnem^ is confined to the mother cell such that their daughter cells have the ability to respond to the mating pheromone. Confinement of Whi3^mnem^ involves its association with the endoplasmic reticulum membranes and the compartmentalization of these membranes by the lateral membrane diffusion barrier at the bud neck, the limit between the mother cell and the bud. However, during the first cell division after the establishment of the pheromone refractory state, this adaptation is more likely to be inherited by the daughter cell than in subsequent cell divisions. Here, we show that the first cell division is associated with larger daughter cells and cytokinesis defects, traits that are not observed in subsequent cell divisions. The cytoskeletal septin protein shows aberrant localisation in these divisions and the septin-dependent endoplasmic reticulum membrane diffusion barrier is weakened. Overall, these data suggest that cytokinesis defects associated with prolonged cell division can alter the confinement and inheritance pattern of a cellular memory.

## 1. Introduction

Single cells have developed strategies to adapt to environmental conditions that can be changing rapidly and drastically. For example, the changes in temperature or salt concentration induce a cellular response to cope with these stresses and promote cell survival [1]. Interestingly, some of these adaptations can be maintained as cellular memories [2,3,4,5,6], in order for cells to respond faster or better to recurrent stresses.

The adaptation to deceptive mating attempts offers such an example of an adaptation that can be stored as a memory by single *Saccharomyces cerevisiae* cells. The haploid cells secrete a mating pheromone that is detected by a plasma membrane receptor. The binding of the pheromone to its receptor activates a mitogen activated protein kinase pathway that ultimately triggers the expression of pheromone responsive genes and a cell cycle arrest in the G1 phase of the cell cycle [7,8]. Cells responding to the pheromone grow towards the source of pheromone in a cellular extension called shmoo. Two haploid cells shmooing towards each other eventually establish a contact and can fuse to form a diploid zygote. However, if no mating partner is in reach, cells can take the decision to ignore the pheromone signal, escape the pheromone induced cell cycle arrest (in short, escape) and therefore resume their cell cycle, despite the presence of the pheromone [9,10]. The escape is maintained for the remainder of the life of the cell that experienced the deceptive mating attempt, hence it is stored as a cellular memory [10]. Remarkably, most of the daughter cells of these experienced mother cells do not inherit the pheromone refractory state and shmoo in the presence of the pheromone.

The molecular mechanism of escape involves the mRNA binding protein Whi3. In addition to its RNA recognition motif that allows Whi3 to bind to a specific set of mRNAs [11,12], the sequence of Whi3 also contains two prion-like domains [13]. These prion-like domains allow Whi3 to change its conformation and its function in response to prolonged exposure to the pheromone. Whi3, in its native form, binds to the mRNA encoding the G1 cyclin Cln3, and inhibits its translation [14]. This inhibition allows cells to respond to the pheromone and arrest the cell cycle in G1 because Cln3 is required to activate the cyclin dependent kinase Cdc28. In response to prolonged pheromone exposure, the conformational change in Whi3 releases this inhibition; *CLN3* mRNA is now expressed and can activate Cdc28 to commit cells to the S phase [10]. Interestingly, the conformational change in Whi3, that depends on both of its prion-like domains, induces its condensation into super-assemblies that are self-templating. Indeed, the Whi3 prion-like domain can form fibrils in vitro and behaves similarly to a prion. However, unlike prions, Whi3 is inherited asymmetrically during cell division. Whi3 super-assemblies are retained in the mother cells and encode the memory of past deceptive mating attempts [10]. The self-templating nature of the Whi3 conformational change allows for the maintenance of the pheromone refractory state over time. In contrast, daughter cells from escaped mother cells are born free of Whi3 super-assemblies and can shmoo in the presence of the pheromone. Therefore, we termed Whi3 a mnemon, to distinguish its asymmetric inheritance during cell division to that of prions, which also rely on a self-templating conformational change but are inherited by daughter cells, hence symmetrically.

An important difference between Whi3 and the most studied prion in yeast, Sup35 (which forms the [*PSI*^+^] prion [15,16,17,18,19]), is that Whi3 is closely associated with the membranes of the endoplasmic reticulum (ER), while the prion form of Sup35 is much less associated [20]. This association plays a role in the retention of the Whi3 mnemon form (Whi3^mnem^) in the mother cell. Indeed, the endoplasmic reticulum is a continuous structure between the mother cell and the bud [21], yet it is compartmentalised by lateral membrane diffusion barriers at the bud neck, the limit between the mother cell and the bud [22]. While ER luminal proteins can diffuse freely between the mother cell and the bud, ER transmembrane proteins and ER membrane associated proteins display a limited diffusion across the bud neck. Barriers have been shown to depend on the function of several proteins, including the septin family of cytoskeletal proteins [23], the GTPase Bud1 or proteins involved in the synthesis of ceramides [21,24]. Defects in any of these proteins result in a strong weakening of the diffusion barrier. Consequently, in cells with a defective diffusion barrier, daughter cells of mother cells that have escaped are more likely to inherit the pheromone refractory state [20]. Since these mother cells do not lose the pheromone refractory state, defects in the barrier formation likely favour the inheritance of Whi3^mnem^ seeds by the daughter cells.

Even though the inheritance of the pheromone refractory state is highly asymmetric, we observed before that during the first division after escape roughly 50% of the first daughter cells do inherit it. This fraction drops below 15% in the subsequent daughter cells [20]. We therefore suspected that the first cell division after escape may often be accompanied by a defective diffusion barrier, a hypothesis that we test here.

## 2. Results

### 2.1. First Daughter Cells from Escaped Mother Cells Are Bigger Than the Subsequent Ones

We noticed that the first daughters emerging from the cells that had escaped the pheromone-induced cell cycle arrest were larger than the subsequent ones. This was confirmed by measuring the area of the 5 first daughter cells after escape. Indeed, the first daughter is significantly larger (35.7 ± 9.4 µm^2^) than the other ones (28.12 ± 5.7 µm^2^, 26.9 ± 5.4 µm^2^, 24.6 ± 4.2 µm^2^ and 23.6 ± 4.1 µm^2^ for the second, third, fourth and fifth daughter cells, respectively, Figure 1A,B). Accordingly, the time between the emergence from the mother cell and cell separation was significantly longer for the first daughter cell than the subsequent ones (Figure 1C).

This difference in cell size seemed to be associated with the fate of the daughter cells. We compared the average cell size of the first daughter cells that shmooed upon birth to that of the first daughter cells that entered the cell division cycle upon birth. Cells that shmooed were significantly smaller (29.0 ± 7.1 µm^2^) than cells that budded (39.0 ± 8.7 µm^2^, *p* < 0.001).

Altogether, these data provide a link between the cell size of the daughter cells and their fate, either shmooing or budding, with the bigger daughter cells more likely to bud upon birth.

### 2.2. First Daughter Cells from Escaped Mother Cells Tend to Remain Attached to Their Mother Cells

When analysing the data above, we observed that the first daughter cells appeared to remain attached to their mother cells after cytokinesis. Daughter cells that were not attached rotated soon after birth (Figure 2A). On the contrary, daughter cells that remained attached did not rotate after birth (Figure 2B), and usually until the end of the movie. In this set of experiments, we found that 74.9 ± 2.6% (3 experiments, n = 103 cells) of the first daughter cells were attached to their mother cells after birth. Except in one instance (1 in 103 cells), second daughter cells were not attached to their mother cells and neither did the subsequent daughter cells. Interestingly, the daughter cells that remained attached were also larger (38.4 ± 8.6 µm^2^) than the ones that did not remain attached to their mother cells (27.6 ± 6.9 µm^2^, *p* < 0.0001, Figure 2C). Similarly to larger first daughter cells being more prone to budding upon birth, cells that remained attached to their mother cells were largely budding upon birth, while all cells that detached from their mother cells always shmooed (Figure 2D). Finally, we compared the escape timing of mother cells that had their first daughter cell shmooing or budding. We found that when the first daughter cell shmooed upon birth, the mothers had escaped earlier (391.0 ± 154.8 min) than the mother that had a first daughter cell budding upon birth (481.9 ± 189.1 min, *p* = 0.021, unpaired two-tailed Welch’s *t*-test, Figure 2E).

Taken together, our data suggest that the first cell division after escape from the pheromone-induced cell cycle arrest is abnormal, characterised by a large cell size of the daughter cell, a defect in cell–cell separation and an increased inheritance in the pheromone refractory state by the first daughter cell compared to the subsequent ones.

### 2.3. The First Cell Division after Escape Is Characterised by Septin Defects

Since the first cell division after escape appeared to display defects in cytokinesis, we next asked whether the septin cytoskeleton localised properly during escape. Septins form a patch during bud emergence that is then transformed into a ring. Upon cytokinesis, the septin ring splits into two rings that disassemble upon cell division completion. We therefore tagged the septin subunit Cdc10 endogenously with a GFP and imaged the cells upon pheromone exposure. Septins were shown to track the pheromone gradient and localise at the shmoo tip in uniform pheromone concentration. Most of the time, we observed the formation of patches of Cdc10 at the shmoo tip, followed by the formation of a bright ring at the bud neck and a subsequent ring splitting event (Figure 3A). Finally, the rings disassembled and a new ring formed at the mother’s second bud neck. However, we observed that in 41.5 ± 14.1% of the cases, the localisation of Cdc10 was altered during the first division after escape. For example, we observed that in addition to the split rings, patches of Cdc10-GFP formed at the bud cortex (Figure 3B), or the formation of aberrant rings with very bright extensions (Figure 3C). Remarkably, most of the 1st daughter cells born from cell divisions that encountered an aberrant septin localisation did bud upon birth, while 1st daughter cells born from a cell division with a normal septin localisation shmooed upon birth (Figure 3D). Interestingly, we did not observe a defect in septin localisation during the second cell division after escape.

Overall, these data suggest that the septin assembly is often defective during escape and that this defect contributes to the inheritance of the pheromone refractory state by the first daughter cells.

### 2.4. Daughter Cells of shs1∆ Cells Tend to Inherit More of the Pheromone Refractory State Than Wild-Type Cells

Since the first daughter cells were more likely to inherit the pheromone refractory state and that septin assembly appeared defective during the first cell division after escape, we next asked if septins were required for the asymmetric inheritance of the pheromone refractory state. We had previously observed that diffusion barrier defective cells (*sur2∆, bud1∆ or bud6∆* cells, [20]) were more likely to inherit the pheromone refractory state. We exposed wild-type cells and septin defective cells (cells deleted for the gene encoding the Shs1 septin subunit—*shs1*∆ mutant cells) to 7 nM of the pheromone and imaged them for 16 h using time-lapse microscopy (Figure 4A). Wild-type cells were more likely to establish the pheromone refractory state and therefore escape within 16 h than *shs1*∆ cells (91.5% of wild-type cells escaped within 16 h compared to 69.2% of *shs1*∆ mutant cells, Figure 4B). We next analysed the inheritance of the pheromone refractory state by the daughter cells of the mother cells that had escaped. Overall, we found that more of the 1st and 2nd *shs1*∆ daughter cells inherited the pheromone refractory state, (66.9 ± 6.3% and 30.4 ± 12.3% for *shs1*∆ 1st and 2nd daughter cells, respectively, compared to 46.5 ± 4.5% and 12.2 ± 3.3% for wild-type 1st and 2nd daughter cells, respectively, *p*-value = 0.0062 and 0.0169 for 1st and 2nd daughter cells, respectively, obtained from a two-way ANOVA, Figure 4C).

Thus, we conclude that septins are required for the retention of the pheromone refractory state in the mother cell.

### 2.5. The Endoplasmic Reticulum Lateral Membrane Diffusion Barrier Is Weakened during the First Division after Escape

Because there seemed to be a septin defect during the first division after escape and that septins are required for the retention of the pheromone refractory state in the mother cell, we asked whether the ER diffusion barrier was somehow impaired during the first division after escape. To test the strength of the diffusion barrier we performed fluorescence loss in photobleaching (FLIP) experiments in cells expressing the ER transmembrane protein Sec61 fused to a GFP. Before each image of the movie, we photobleached an area in the mother cell and measured the fluorescence intensity decay in the mother part and in the bud part, in cells that had not been exposed to the pheromone (Figure 5A,B) and in cells that had been exposed to the pheromone for 5 h (Figure 5C,D). From these data, we extracted the time it took for each compartment to reach 70% of its initial value and calculated the barrier index for each cell (the barrier index is defined by the time it takes for fluorescence to decay to 70% in the bud part divided by the time it takes for fluorescence to decay to 70% in the mother part). From these experiments, we observed that the average barrier index was higher in the untreated cells compared to cells treated with the pheromone (*p*-value = 0.0015), yet the difference was small (there was an average barrier index of 12.86 ± 6.1 in untreated cells and of 10.80 ± 9.4 in cells treated with the pheromone, Figure 5E). However, we noticed that for many cells that had escaped pheromone arrest, the barrier index was very low. Indeed 30% (15 out of 50) of the cells treated with the pheromone displayed a lower barrier index that was lower than the lowest barrier index of the untreated cells. Therefore, we conclude that during the first division after escape, the ER diffusion barrier is impaired.

## 3. Discussion

Single cells have the ability to store information of their past history to adapt to changing environments. The pheromone refractory state encoded by the Whi3^mnem^ provides cells with a memory that is largely kept by the mother cell during cell division. The advantage of this retention is that newborn daughter cells retain the ability to respond to the pheromone and mate, should a mating partner be close enough. Therefore, budding yeast cells have evolved a mechanism to maintain the asymmetric inheritance of the self-replicating Whi3^mnem^ [20].

Whi3 binds to the membranes of the ER through a yet unknown tether. This interaction is however important because the membranes of the ER are compartmentalized. Although continuous between the mother cell and the bud, the lateral membrane diffusion barriers establish a boundary at the bud neck. The diffusion of the ER membrane associated proteins across the bud neck is limited by the diffusion barrier. Here, we have observed that during the first division after escape, the pheromone refractory state is much more inherited than in the next divisions. Our data argue that this phenomenon is linked to several aspects. Firstly, during the first division after escape, the bud is much larger than in the next ones. Therefore, the duration of the cell division is increased, which allows for a longer time window for Whi3^mnem^ to cross the bud neck. Indeed, even though diffusion barriers are formed, they introduce a bias and not a complete block of diffusion. We note however that in the case of cells escaping the pheromone-induced cell cycle arrest, the mother compartment is very large (as they shmooed for several hours). Secondly, we found that septins, which are required for the formation of diffusion barriers [21,22], mislocalise during the first division after escape. Third, we found that the ER diffusion barrier is not as strong after escape as in the cells that undergo an unperturbed cell cycle. Altogether, we propose that a long first cell division, combined with an impaired diffusion barrier, allow for the inheritance of Whi3^mnem^ and the pheromone refractory state by the first daughter cells.

What is inducing the mislocalisation of septins and a long cell cycle just after escape remains unclear. A hypothesis that we can put forward is that cells that have shmooed for several hours, and reached a very large cell size, are experiencing some sort of stress that signals to the septin cytoskeleton. In support of this idea, the cytokinetic septin ring does not disassemble properly in strong ER stress conditions [25] and septins also mislocalise during strong ethanol stress [26]. However, not all stresses seem to affect septin localisation, because the ER diffusion barrier is functional in cells exposed to heat stress [27].

If the large cell size experienced by cells that have shmooed is indeed sensed and triggers septin mislocalisation as well as diffusion barrier weakening, similar phenotypes may be conserved in other organisms. In rodents, neural stem cells establish ER diffusion barriers at the cleavage plane during the cell division [28]. However, during aging, the diffusion barrier weakens, leading to more symmetric cell division. Interestingly, cellular senescence is associated with cellular enlargement [29]. During aging, hematopoietic stem cells become large and their functions decline [30]. It is therefore possible that stem cell enlargement induces defects in the diffusion barriers that could impact the function of these cells and affect the aging process.

## 4. Materials and Methods

### 4.1. Yeast Strains

The strains used for escape from the pheromone arrest were derivatives of the s288c BY4743 wild-type (yFC05: *MAT***a**, *his3∆1 leu2∆0, ura3∆0 met15∆0 lys2∆0 ADE2 TRP1 bar1::kanMX*) with deletion of *SHS1,* and GFP tagging of Cdc10 and Sec61 obtained according to Janke et al. [31] (yFC40: *CDC10*-*GFP:HIS3;* yFC44: *shs1::kanMX* and yFC67: *SEC61*-*GFP:TRP1*) using the following primers:

Primer sequences to delete *SHS1*:

*Shs1*-*S1*: AGAGCCCCAAAGATCTGCTTATAATTGCTAGAAAAATATATTATTAATCATGCGTACGCTGCAGGTCGAC.

*Shs1*-*S2*: ATTTATTTATTTATTTATTTGCTCAGCTTTGGATTTTGTACAGATACAACTCAATCGATGAATTCGAGCTCG.

The deletion of SHS1 was checked by PCR genotyping using the following primers for the presence of the wild-type *SHS1* gene:

*Shs1*-*up*: *TGCTGATAACTTGGAAGCATC* and *Shs1*-*cig*: *TCTACGAAATAAGTTAATTGG*. To verify the correct insertion of the deletion cassette, we used the primers: *Shs1*-*up TGCTGATAACTTGGAAGCATC* and *S1*-*reverse GTCGACCTGCAGCGTACG*.

Primer sequences to tag Cdc10 with GFP:

*Cdc10*-*S2*: ATGCGAATAGTCGTTCCTCAGCTCATATGTCTAGCAACGCCATTCAACGTATCGATGAATTCGAGCTCG.

*Cdc10*-*S3:* AATTCTTAATAACATAAGATATATAATCACCACCATTCTTATGAGATTCACGTACGCTGCAGGTCGAC.

Primer sequences to tag Sec61 with GFP:

*Sec61*-*S2:* GCTAAATGCGATTTTTTTTTTCTTTGGATATTATTTTCATTTTATATTCAATCGATGAATTCGAGCTCG.

*Sec61*-*S3:* GGAAGGTGGGTTTACTAAGAACCTCGTTCCAGGATTTTCTGATTTGATGCGTACGCTGCAGGTCGAC.

Cells were grown in a yeast extract and peptone (Formedium, Norfolk, UK, CCM0410) broth supplemented with 2% dextrose (Formedium, Norfolk, UK, GLU03) for escape experiments and Synthetic Complete media with 2% dextrose (Formedium, Norfolk, UK, CSC0210) for fluorescence microscopy experiments. In all cases, cells were grown at 30 °C in a shaking incubator (180 rpm) overnight and diluted in the morning to OD 0.15 to reach mid-log phase (OD 0.5 to 0.6) for the experiments.

### 4.2. Microscopy

All images except for FLIP experiments were acquired on a Deltavision Elite (GE Healthcare) equipped with a sCMOS camera and solid-state light-emitting diodes controlled by the software Softworx. A fluorescein isothiocyanate filter was used for imaging GFP fluorescence. The time lapse was set at 15 min for 16 h for escape experiments and 20 min for 16 h for imaging GFP fluorescence.

Experiments were carried out with the ONIX microfluidic perfusion platform with Y04C microfluidic plates (CellAsic, Merck, Darmstadt, Germany), with a flow rate set at 3 psi. The medium was a yeast extract peptone dextrose (YPD) supplemented with 20 mg/mL casein and containing 7 nM alpha-factor. Alpha-factor (Sigma-Aldrich, St. Louis, MO, USA, T6901) was aliquoted in 10 µL samples at 1 mg/mL in water supplemented with 20 mg/mL casein and defrosted aliquots were used only once. For GFP imaging, cells were grown in an SC medium supplemented with 20 mg/mL casein and containing 9 nM alpha-factor.

### 4.3. Quantification of Shmooing

Shmooing or budding states were inspected visually. During microfluidic experiments, images were taken every 15 min. Unbudded cells that showed a polarized growth were counted as shmooing. Unbudded cells undergoing isotropic growth were counted as G1 cells. Usually, these cells soon started forming a bud.

### 4.4. Cell Sizes Measurements

The areas of the daughter cells were determined using FIJI by tracing the contour of the cell and extracting the area of the contour.

### 4.5. Quantification of Cell Division Timing

Using FIJI, the time between bud emergence and cell separation (the formation of a visible dark line on the microscopic images at the bud neck reported on the formation of the septum at the bud neck) was measured for the first five daughter cells of escaped mother cells. Most of the time, the following bud emerged in the time frame after cell separation occurred.

### 4.6. FLIP

Overall, we followed the detailed steps described by Bolognesi et al. [32]. FLIP experiments were performed using a confocal microscope Zeiss LSM980 NLO with a 60x Plan Apo (NA 1.4) oil immersion objective, operated by the Zeiss Zen Blue software. Cells were grown on YPD plates (yeast, peptone and 2% dextrose) at 30 °C and resuspended in synthetic complete medium. Cells were then placed on a synthetic complete agar pad and imaged in an 8-well chamber (Ibidi GmbH, Gräfelfing, Germany). A total of 1.6% of the laser power was used to image the cells and 80% of the laser power was used for photobleaching (100 iterations in the bleached region). Quantification was performed using FIJI [33] and curve fitting using Graphpad Prism 9.5.0.

### 4.7. Statistics

Graphs and statistics were performed using Graphpad Prism 9.5.0.

## Figures and Tables

**Figure 1 ijms-24-03003-f001:**
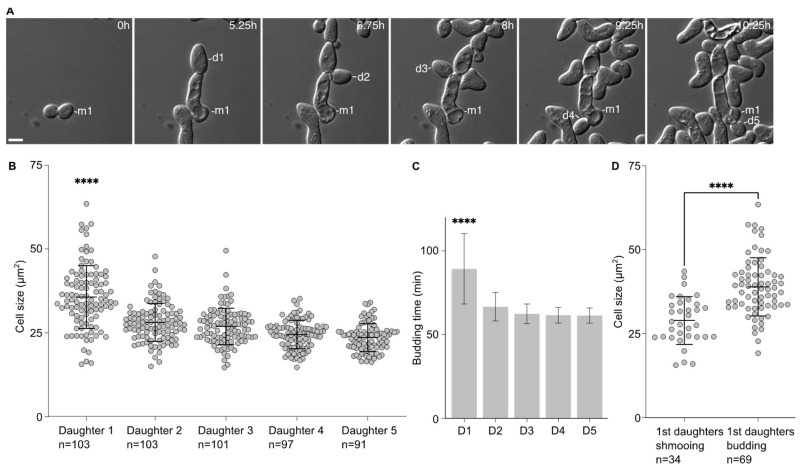
The first daughter cells are larger than the subsequent ones after escape. (**A**) Escape of a WT cell exposed to 7 nM of pheromone. M1 = mother cell. D# = daughter cells of the escaped mother cell. Scale bar = 5 µm. (**B**) Cell size (area; average ± SD) in µm^2^ of the 1st and subsequent daughter cells. The variable n = number of cells obtained from 3 independent experiments. The **** = *p* value < 0.0001 obtained from an ordinary one-way ANOVA comparing the cell size of the 1st daughter to the size of any of the subsequent daughter cells. (**C**) Time for each daughter cell from bud emergence from the mother cell to cell separation (average ±SD). The **** = *p* value < 0.0001 obtained from an ordinary one-way ANOVA comparing the timing of the 1st daughter to the timing of any of the subsequent daughter cells. The variable n as in A. (**D**) Cell size (area; average ± SD) in µm^2^ of the 1st daughter cells that are either shmooing or budding upon birth. The variable n = number of cells obtained from 3 independent experiments. The **** = *p* value < 0.0001 obtained from an unpaired two-tailed *t*-test.

**Figure 2 ijms-24-03003-f002:**
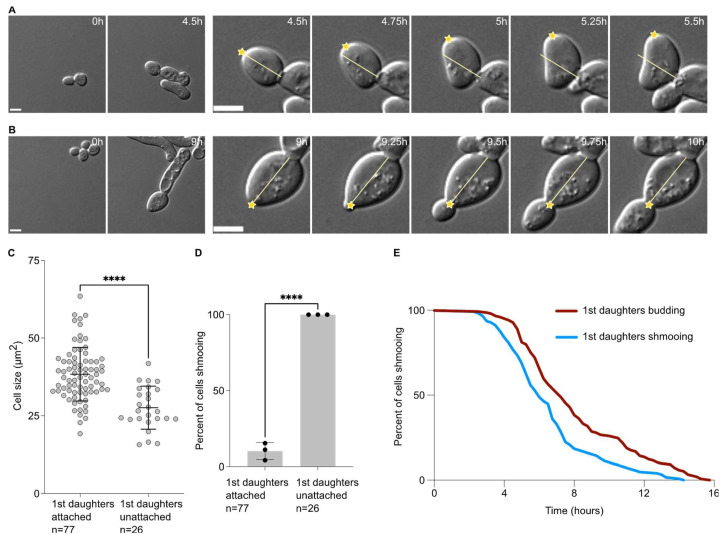
Attached first daughter cells are larger and more likely to bud upon birth than unattached ones. (**A**) Escape of a WT cell exposed to 7 nM of pheromone with a first daughter cell unattached. Left panel, pictures of the mother cell until the bud separates from the mother cell. Right panels, focus on the first daughter cells during the first hour after cytokinesis. The yellow line indicates the long axis on the daughter cell at the cytokinesis point. The yellow star indicates the daughter tip. Note that in this case the yellow star deviates from the yellow bar, showing that the first daughter cell is free to move. Scale bar = 5 µm. (**B**) Same as A for an attached first daughter cell. Note that the yellow star does not deviate from the yellow bar, showing that the first daughter cell is not free to move. Scale bar = 5 µm. (**C**) Cell size (area) in µm^2^ of the 1st daughter cells, either attached or unattached to their mother cell. The variable n = number of cells obtained from 3 independent experiments. The **** = *p* value < 0.0001 obtained from an unpaired two-tailed *t*-test. (**D**) Percent of cells shmooing, whether they are attached or unattached to their mother cells upon birth. The variable n = number of cells obtained from 3 independent experiments. The **** = *p* value < 0.0001 obtained from an unpaired two-tailed *t*-test. (**E**) Percentage of initial cells still shmooing after the indicated time. The variable n > 154 cells.

**Figure 3 ijms-24-03003-f003:**
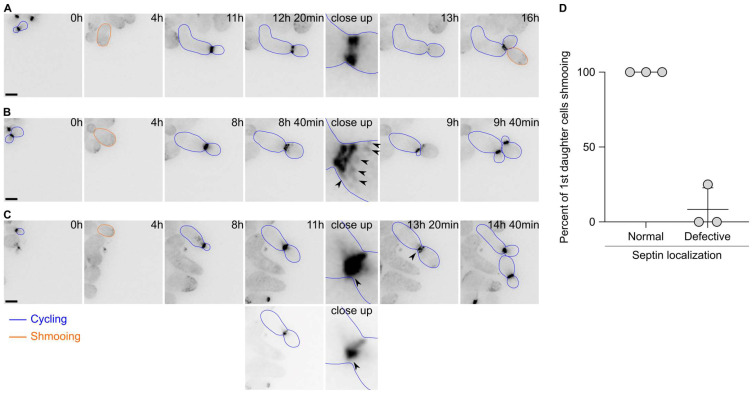
Septin localisation is often aberrant during the first cell division after escape. (**A**) Images of a wild-type cell expressing Cdc10-GFP and displaying a normal localisation during escape. Note that the 1st daughter cell is shmooing upon birth. Scale bar = 5 µm. (**B**) Same as A except that the Cdc10-GFP signal shows strong patches (black arrowheads) on the bud side and that the 1st daughter cell is budding upon birth. Scale bar = 5 µm. (**C**) Same as B except that the Cdc10-GFP signal shows a bright accumulation (black arrowhead) at the bud neck. The 1st daughter cell is budding upon birth. The two lower panels depict an adjusted contrast of the same images compared to all other images to better visualise the Cdc10-GFP accumulation. Scale bar = 5 µm. (**D**) Quantification of the fate of the 1st daughter cells after escape depending on the septin localisation. Average ± SD. N = 3 experiments, with > 60 cells. The *p*-value = 0.0082 using a Welch’s *t*-test.

**Figure 4 ijms-24-03003-f004:**
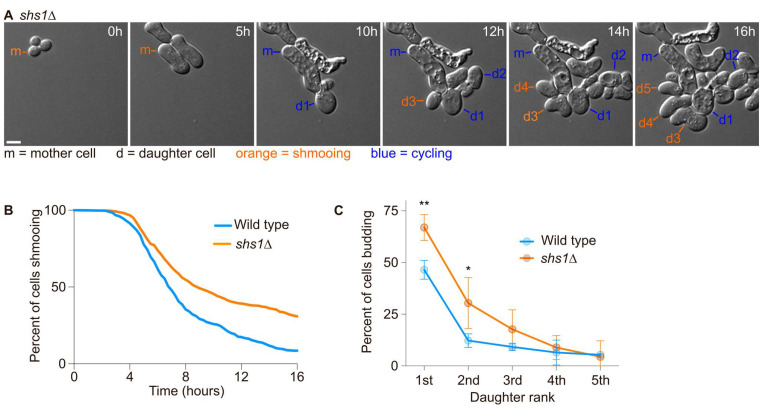
The septin subunit Shs1 is required for an efficient escape and asymmetric inheritance of the pheromone refractory state. (**A**) Escape of a WT cell exposed to 7 nM of pheromone. Here, m = mother cell, d# = daughter cells of the escaped mother cell. Scale bar = 5 µm. (**B**) Percentage of initial cells still shmooing after the indicated time. N > 305 cells. (**C**) Percentage of the daughter cells budding immediately after separation from the mother cell. Mean ± SD are presented from 3 independent experiments. The *p*-values ** < 0.01, * < 0.05 obtained from a two-way ANOVA.

**Figure 5 ijms-24-03003-f005:**
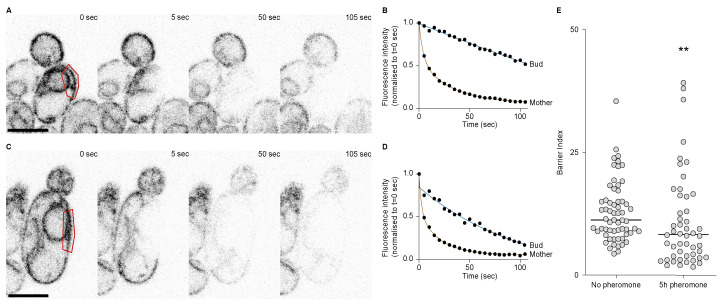
The ER diffusion barrier is weakened during the 1st division after escape. (**A**) FLIP of Sec61-GFP in an untreated wild-type cell. Bleaching area in red. Scale bar = 5 µm. (**B**) Fluorescence decay over time in the mother and bud parts of the cell in (**A**). (**C**) FLIP of Sec61-GFP in a wild-type cell treated with the pheromone for 5 h. Scale bar = 5 µm. (**D**) Fluorescence decay over time in the mother and bud parts of the cell in (**C**). (**E**) Barrier indices for cells untreated (n = 61 cells) or treated with the pheromone for 5 hours (n = 50 cells). The ** *p* value = 0.0015 obtained from a Mann–Whitney test.

## Data Availability

Materials and data are available upon request.

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
