# Peer review of "Septin Defects Favour Symmetric Inheritance of the Budding Yeast Deceptive Courtship Memory"

_ijms, 2023, doi:10.3390/ijms24033003_

Round 1
Reviewer 1 Report
The ms describes fine cell biology experiments that show that cellular memory based on a prion-like mnenon, that fades over generations in yeast, are stronger in the first daughter cell because of defects at the bud neck. The process relies on septins. These results are obtained through expert cell biology experiments. Conclusions are sound and the discussion relevant. A few typos can be found, suche as an upside down "A " in fig 3A (on my copy at least) and a "state" missing in "refractory at the end of the sentence on line 175.
Author Response
We would like to thank this reviewer for their support.
We have added the missing word “state” on line 175 (now line 178).
We suppose that the “upside down ‘A’” in fig 3A came from the converted document on the journal website, because in our source files, the A is normally written.
Reviewer 2 Report
The scope of this work resides on the study of the molecular process that governs adaptation to mating pheromone and how this is inherited to daughter cells. Authors describe some phenotipic traits that differentiate 1st daughter cell from subsequent divisions after sexual pheromone treatment. The main findings show that the first daughter cell is larger in size and presents septin mis-localization at the bud neck which affects pheromone refractory behavior.
This paper is a continuation of a previous paper published by the same group (Current Biology 32:963, 2022), where the role of mnemon prion-like Whi3 protein was described. In fact, this previous paper has greatly influenced the content of the present manuscript. Although the results described here are interesting, the conclusions raised are based, in part, on the Curr Biol paper. This is true for the abstract, which refers greatly to the role of Whi3mnemon; the introduction which is focused mainly in this same protein; and in some key conclusions written in the discussion section, which were not measured in this MS, for example, the conclusions written in page 10 line 239, and page 11 line 248.
Minor comments:
Materials and methods can be less condensed and more descriptive.
It is not easy to understand the conclusion raised in the abstract which states that stress can alter the confinement and inheritance pattern of a cellular memory. What kind of stress are authors referring to?, does this mean that 1st generation cells are subjected to different stress than second (and so on) generation cells? I think that this should be clarified and based on experimental data.
Author Response
We would like to thank this reviewer for their support.
We have now added the quantification of the duration of cell division (from bud emergence to cell separation) and show that indeed the duration of cell division is longer for the 1st daughter cell after escape from pheromone arrest than for the subsequent ones. This has been added in the text and also as Figure 1C.
The materials and methods section has been expanded.
Concerning the stress that may be experienced only during the first division after escape, we have removed the sentence from the abstract. This will be the topic of a full article in the near future and we agree that this should not be mentioned in this abstract as there is no data backing it.
Reviewer 3 Report
The paper present data on the role of septin and ER diffusion barrier during the escape pheromone induced cell cycle in budding yeast. The research contributes to the better understanding of how cellular memory might be inherited in Eukaryotes, therefore the topic is up to date and might attract the attention of other researchers not only yeast fans’.
However, there are some issues which should be answered/corrected before publication:
Questions:
1. The Authors demonstrated that “septin localization is often aberrant during the first cell division after escape” (Figure 3B and C). According to the “close up” images, the mispositioning of septin seems to be altered in the two cells, although the experimental conditions were identical. How can it be explained?
2. Figure 3D shows that wild type cells having defective septin localization practically do not shmoo. Figure 4D shows that septin mutants shmoo more efficiently than wild type cells. Does it mean that mislocalization of the septin ring cause “stronger” phenotype then a subunit deletion?
3. The Authors claim that “Daughter cells of shs1∆ cells tend to inherit more of the pheromone refractory state than wild type cells.” (lanes 170-71). Figure 4B shows the opposite: higher percent of the mutant cells shmoo than that of wild type cells, which means for me that less mutant cells inherited the mother’s pheromone refractoy state. Please, explain this paradox.
4. The Authors did not perform any stress experiments, therefore the last sentence (lanes 27-28) of the Abstract might be misleading and should be omitted.
Minor issues:
The Materials and methods chapter seems to be a bit overlooked:
5. I suggest Yeast strains and growth conditions paragraph first, where besides strain descriptions (e.g. genotypes, reference/origin), the culture condions are descibed according to the international standards [Media, growth temperature, cultures used for experiments (e.g. early-/mid-log phase or cell number/ml)].
6. The Authors produced two GFP-tagged constructions, therefore primers used for PCR reactions should also be listed.
7. The Authors used microfluidic technic (lane 293) therefore the experimenal conditions should be described (device properties, flow rate, etc.)
8. As regards to FLIP experiments the approximate positions of the photobleached areas should be labelled at least on Figure 7, instead of the “... we photobleached an area…” (lane 199) description.
Figures:
Figure 1A: The characters intended to label the cells are too large and should be placed above/beside/below the cells or other labels should be used which do not cover the area of interest.
The figure and the legend is superimposed.
Figure 3A: The label “A” is upside down. C: 0h: only the half of the cell is outlined.
Please describe what red and green outlines mean.
Figure 4A: See note at Figure 1A.
Figure 5. label “A” is missing.
Author Response
We would like to thank this reviewer for their support. We respond below to their suggestions and questions point by point.
- In these experiments, we observed that septin localization was indeed aberrant in roughly 41% of the cells, during the first division after escape. It is unclear to us why not all cells display a septin mislocalization. We can envision several possibilities: 1) using conventional fluorescence microscopy, and during time lapse experiments for which we cannot expose the cells to too much light without introducing a stress, it is possible that we cannot detect all septin defects, 2) Since the amount of cells that display a defect is roughly 41%, it is possible that either cells that have divided before responding to pheromone (±50% of an exponentially growing population) or on the opposite, only virgin cells (±50% of an exponentially growing population) do have septin assembly defects upon escape from pheromone arrest, 3) As for many biological processes, it is possible that many cells do experience the process of escaping differently. We have observed that cells responding for a long time to pheromone are in fact activating signalling pathways to respond to a change in plasma membrane tension. However, this change in plasma membrane tension is not exactly the same for all cells, which may have grown in size differently (some are more curved than others, some are longer than others, etc…). Therefore, we think that the signalling pathways (downstream of TORC2) triggered by the change in plasma membrane tension affect septin dynamics. Indeed, the Piatti group has shown that Pkc1 affects septin dynamics (PMID: 26179915).
Overall, we have not yet a definitive answer to this question, but hope that this will be the case in the near future.
- The reviewer points out that septin mislocalization seems to have a stronger defect than deletion of a septin subunit. This is an interesting point that we have also noticed. The reason may lie in the different subunit that we used. Cdc10 and Shs1 are two subunits that are both required for the formation of a functional diffusion barrier. However, they do not engage in the exact same septin filaments. Cdc10 is the central subunit in the septin filaments and is therefore present in all septin assemblies. As such, CDC10 is essential in many yeast backgrounds. Shs1 is localized at the ends of the septin filaments and only in a subset, because another subunit, Cdc11, can replace it. Therefore, it is absolutely possible that the deletion of SHS1 has a weaker effect than the mislocalization of Cdc10. Alternatively, the difference may come from the fact that shs1∆ cells are generally larger and can have complex shapes in response to pheromone. If the shs1∆ mother cells are dramatically larger than the WT mother cells, it may be possible that the likelihood of seeds of Whi3mnem to diffuse to daughter cells is decreased. We are developing tools to visualise and quantify seeds of Whi3mnem and hope to be able to answer this question in the near future.
- Figure 4B displays the number of initial cells still shmooing over time. In other words, the curves describe the escape timing of the mother cells before they escape. In these quantifications, the daughter cells born from mother cells that have escaped are not included in order for us to distinguish between the timing required for cells to establish the pheromone refractory state and the inheritance of the pheromone refractory state (described in Figure 4C, in which we do observe that shs1∆ daughter cells are more likely to inherit the pheromone refractory state from their mothers).
We have rephrased our sentence line 183 to hopefully better explain this difference.
- Following the suggestion of this reviewer, we have removed the mention of stress in the last sentence of the abstract.
- We have now expanded the materials and methods section according to this reviewer’s suggestions.
- We have included the primer sequences used to create the yeast strains.
- We have now described the microfluidic set up in details.
- We have added the ROI used for the FLIP experiments in Figure 7.
- Figure 1A: The characters intended to label the cells are too large and should be placed above/beside/below the cells or other labels should be used which do not cover the area of interest.
We have moved the characters beside the cells so that the reader can see the whole cell.
The figure and the legend is superimposed.
We suppose this is coming from the file conversion by the journal, as this is not the case in our files. We hope this will be resolved in the next version.
Figure 3A: The label “A” is upside down. C: 0h: only the half of the cell is outlined.
Please describe what red and green outlines mean.
We suppose this is coming from the file conversion by the journal, as this is not the case in our files. We hope this will be resolved in the next version.
We have changed the colours to orange and blue, to have the same colour code in the figures 3 and 4 and described their meaning on the figure.
Figure 4A: See note at Figure 1A.
We have moved the characters beside the cells so that the reader can see the whole cell.
Figure 5. label “A” is missing.
We suppose this is coming from the file conversion by the journal, as this is not the case in our files. We hope this will be resolved in the next version.